# Disrupting Neurons and Glial Cells Oneness in the Brain—The Possible Causal Role of Herpes Simplex Virus Type 1 (HSV-1) in Alzheimer’s Disease

**DOI:** 10.3390/ijms23010242

**Published:** 2021-12-27

**Authors:** Matylda Barbara Mielcarska, Katarzyna Skowrońska, Zbigniew Wyżewski, Felix Ngosa Toka

**Affiliations:** 1Department of Preclinical Sciences, Institute of Veterinary Sciences, Warsaw University of Life Sciences–SGGW, Jana Ciszewskiego 8, 02-786 Warsaw, Poland; ftoka@rossvet.edu.kn; 2Department of Neurotoxicology, Mossakowski Medical Research Institute, Polish Academy of Sciences, Adolfa Pawińskiego 5, 02-106 Warsaw, Poland; kskowronska@imdik.pan.pl; 3Institute of Biological Sciences, Cardinal Stefan Wyszyński University in Warsaw, Dewajtis 5, 01-815 Warsaw, Poland; zbigniew.wyzewski@gmail.com; 4Department of Biomedical Sciences, Ross University School of Veterinary Medicine, Basseterre 42123, Saint Kitts and Nevis

**Keywords:** Alzheimer’s disease, herpes simplex virus type 1 (HSV-1), herpes simplex encephalitis, brain, astrocytes, microglia, oligodendrocytes

## Abstract

Current data strongly suggest herpes simplex virus type 1 (HSV-1) infection in the brain as a contributing factor to Alzheimer’s disease (AD). The consequences of HSV-1 brain infection are multilateral, not only are neurons and glial cells damaged, but modifications also occur in their environment, preventing the transmission of signals and fulfillment of homeostatic and immune functions, which can greatly contribute to the development of disease. In this review, we discuss the pathological alterations in the central nervous system (CNS) cells that occur, following HSV-1 infection. We describe the changes in neurons, astrocytes, microglia, and oligodendrocytes related to the production of inflammatory factors, transition of glial cells into a reactive state, oxidative damage, Aβ secretion, tau hyperphosphorylation, apoptosis, and autophagy. Further, HSV-1 infection can affect processes observed during brain aging, and advanced age favors HSV-1 reactivation as well as the entry of the virus into the brain. The host activates pattern recognition receptors (PRRs) for an effective antiviral response during HSV-1 brain infection, which primarily engages type I interferons (IFNs). Future studies regarding the influence of innate immune deficits on AD development, as well as supporting the neuroprotective properties of glial cells, would reveal valuable information on how to harness cytotoxic inflammatory milieu to counter AD initiation and progression.

## 1. Introduction

Active herpes simplex virus type 1 (HSV-1) infection generates compound biochemical and morphological changes, leading to the injury of neurons and other brain cells that culminate in cell death. Nevertheless, the virus aims to intercept cell machinery to produce viral components. As mentioned by Itzhaki, R. [1], Alzheimer’s disease (AD) is considered as a multifactorial ailment, partly exerted by genetic or environmental factors; however, the neuropathological processes that lead to the disease are still not fully understood. In recent years, many studies advocate the relationship between HSV-1 infection and various neuropsychiatric and neurodegenerative diseases [2,3].

AD is the paramount cause of a decline in cognitive ability [4] and accounts for about 70% of dementia cases [5]. Currently, there is no effective treatment, and a rapidly aging society implies AD as one of the major public health concerns. Pathological grounds and, concomitantly, the hallmark features of the disease comprise deposits of the extracellular misfolded amyloid beta (Aβ) in the form of Aβ plaques, the presence of intracellular neurofibrillary tangles (NFTs) composed of hyperphosphorylated tau, as well as the neuroinflammation, gliosis, and activation of glial cells in the brain [6,7,8].

The role of HSV-1 infection in the brain in AD pathogenesis was originally proposed, several decades ago, in the pioneering works by Ball [9] and Gannicliffe et al. [10]. HSV-1 is a neurotropic, double-stranded (ds) DNA (dsDNA) virus that infects the peripheral sensory neurons and establishes life-long latency in the trigeminal ganglion (TG) [11]. Latent HSV-1 sporadically reactivates and can migrate into the trigeminal nuclei located in the brainstem reaching the thalamus and sensory cortex, resulting in devastating viral encephalitis (herpes simplex encephalitis, HSE) or persisting latent infection in the central nervous system (CNS) [12,13]. Recently published research indicates HSE and persistent HSV-1 infections in the brain as factors that predictably increase the risk or determine the AD origination [14,15,16,17,18]. In addition to causing severe encephalitis, HSV-1 and HSV-2 can also reach the brain without evident clinical symptoms [15], and the subclinical chronic reactivation of latent HSV-1 in the brain is considered to promote AD pathogenesis and accelerate the disease progression [14,15]. Astoundingly, among patients who underwent HSE, an immune reaction in the brain prolonged for up to ten years following the acute onset of the disease, and viral DNA was found in their brains [19]. Evidence of an association of HSV-1 brain infection with AD relates to the features of AD, epidemiology and pathology [20]. The frontal and temporal cortices brain regions affected during HSE are also primarily affected in AD, and the main symptoms of AD often appear in HSE survivors [21]. Research also shows that the combination of HSV-1 infection in the brain and the genetic factor, such as the carriage of an apolipoprotein E ε4 (*APOE-ε4*) allele, is a potent risk factor for AD [22]. Furthermore, over 230 different autosomal dominant pathogenic variants of amyloid precursor protein (*APP*), presenilin 1 (*PSEN1*), and *PSEN2*, which likely increase the risk of AD development, were identified [23]. Mutations of 19 other genes (involved in brain development, cytoskeletal organization, and immune function), which slightly increase the risk of AD development, were identified in the last decade, as reviewed by Bird [24]. Nevertheless, while ~25% of all AD cases are familial, ~75% of AD cases remain with an unknown disease background [24].

## 2. The Active and Latent HSV-1 Infection, and Antiviral Immune Response

Following the infection of a naïve person, HSV-1 establishes latency in the peripheral sensory neurons of the TG or dorsal root ganglia (DRG) as well as vestibular, and facial ganglia [25]. With increasing age, latent HSV-1 infection in the TG is observed in a growing number of people. Nearly 70% of people are seropositive for HSV-1 [26], and infection estimates indicate that around 90% of the human population may be infected with the virus, which underlines its prevalence in the world [27]. Latent HSV-1 can reactivate following damage to the tissue innervated by latently infected neurons, other infections, hormonal imbalance, fever, exposition to systemic physical or emotional stress, and UV light, as reviewed by Stoeger and Adler [28]. Most often, reactivation is clinically asymptomatic or causes cranial nerve disorder in the form of herpes labialis. Under certain circumstances, HSV-1 can infiltrate various parts of the brain, such as the olfactory bulb, temporal lobes, and other regions in which the genetic material of the virus is found; the epithelial surface of the cornea; or cause disseminated infection pertaining to multiple organs, such as the liver, lungs, adrenal glands, and also the brain [29]—HSV infection is the most common cause of severe and life-threatening (approximately 70% of untreated cases are fatal [30]) sporadic encephalitis—HSE, blinding keratitis, and neonatal disseminated herpes [25,31,32].

Primary infections with herpes simplex viruses induce rapid immune response aimed at the efficient inhibition and clearance of HSV-1 infection [33], also preventing virus entry into the brain. During active infection, HSV constantly replicates and produces pathogen-associated molecular patterns (PAMPs), such as viral proteins, DNA, and RNA, as well as cell injury and death-associated products (damage-associated molecular patterns, DAMPs) that activate the host’s pattern recognition receptors (PRRs) and initiate innate immune responses. In particular, during HSV-1 replication, dsRNA is formed as an intermediate that activates toll-like receptor 3 (TLR3), retinoic acid-inducible gene I (RIG-I), melanoma differentiation-associated protein 5 (MDA5), and dsRNA-dependent protein kinase R (PKR) (it is debated whether PKR recognizes HSV-1 RNA or HSV-1–infected host’s genome). The viral glycoproteins gH and gL stimulate TLR2; soluble gD activates the herpes virus entry mediator (HVEM). HSV-1 DNA is sensed by TLR9, interferon γ (IFN-γ)-inducible protein 16 (IFI16), DNA-dependent activator of IFN-regulatory factors (DAI), and cyclic guanosine monophosphate-adenosine monophosphate synthase (cGAS) (Figure 1) (it is speculated whether cGAS recognizes HSV-1 DNA or the host’s mitochondrial DNA released following infection [34]), as reviewed by Danastas et al. and Zhao et al. [35,36].

Innate immunity principally relies on type I interferons (IFNs, IFN-α/β) production, the host’s first line of defense against HSV-1. By binding to corresponding receptors and activating IFN-stimulated genes (ISGs), type I IFNs induce an antiviral response across distinct cell types, such as monocytes, neutrophils, dendritic cells (DCs), macrophages, and natural killer (NK) cells [33], and mediate adaptive immune response, including humoral and cellular components managing antiviral control and latency [37,38].

During latency in neurons, HSV genome is transcribed, latency-associated transcripts (LATs) comprising two major LAT RNA species and several viral microRNAs (miRNAs) are highly expressed [39,40,41], while viral immediate-early genes, e.g., *ICP4*, are expressed at a low frequency; however, the ICP4 antigen can be present in latently infected ganglion neurons [42]. HSV-1 latency is an active process, in which LATs are involved in modulating viral gene expression [40], miRNAs attenuate transcripts of productive infection genes [41], while HSV virions are not detectable. Moreover, LATs downregulate components of type I IFNs pathway [43], but provoke the increased expression of IFN-γ, tumor necrosis factor α (TNFα), IFN-γ-inducible protein 10 (IP-10), and C–C motif chemokine ligand 5 (CCL5) (RANTES), and thereby interpose between the infiltration of the lymphocytic cells as well as chronic inflammation [44]. Laboratory animals with a latent HSV-1 infection in the nervous system, exhibit increased levels of inducible nitric oxide synthetase and cytokines that decrease following the treatment of animals with acyclovir [45]. Such a coherence engenders the belief that HSV-1 latency alone can entail low-grade injury of the CNS cells, without leading to clinical manifestation [45].

Although epithelial cells and neurons are the major target cells during primary and recurrent HSV-1 infection [46], in the brain the virus also infects glial cells, including microglia, astrocytes, and oligodendrocytes [47,48,49,50,51,52]. Glial cells pave the CNS and adopt various important functions [53]. They significantly outnumber neurons and express more selective receptors for HSV-1 than neurons, which results in higher HSV-1 adsorption [54,55]. It has been reported that herpes viruses infect and replicate in astrocytes [56,57], which determine neuronal vulnerability to HSV-1 [58]; oligodendrocytes, the myelin-producing cells, are susceptible to HSV-1 infection in vivo [59,60,61,62] and in vitro [63], and HSV-1 infects microglia that isolate virus-infected neurons in the brain [64].

Microglia, astrocytes, and oligodendrocytes are potent immune cells that protect the brain from pathogens; however, their activation can lead to the weakening of their neuroprotective and homeostatic properties, and the development of a neurotoxic pro-inflammatory environment [65]. While the intricacy of AD stows difficulties in understanding the exact cause and effect of a particular molecular or cellular pathway, research aiming to unravel and understand how to preserve or restore the functions of glial cells can delineate novel therapies to safeguard the integrity of the CNS.

## 3. Amended Production of Various Inflammatory Factors

During HSV-1 infection, neurons and glial cells produce many inflammatory mediators [66], and the rapid development of inflammation following CNS infection suggests that glial cells play a pivotal role in the initiation and progression of encephalitis. It is believed that inflammation, as well as the lowered clearance of misfolded proteins in the CNS, can play crucial roles in neurodegeneration and cognitive decline [67,68]. Moreover, persistent glial cells-mediated inflammation is considered to be a key contributor to the neurodegenerative processes and cognitive ailments observed in AD [69]. HSV-1 infection has been shown to generate high levels of proinflammatory cytokines and activated IFI16 and NLRP3 inflammasomes, inducing the secretion of interleukin 1β (IL-1β), IL-18, and IL-33, compared to healthy humans [70]. HSV-infected cells can contribute to the progression of AD and lowered synaptic density through secretion of IFN-β, IFN-γ, IL-1β, IL-6, chemokine (C–X–C motif) ligand 8 (CXCL8), TNFα, and TGFβ, the immunosuppressive cytokine [71,72,73,74], for which up-regulation is observed both in human AD brain samples, and in transgenic murine models of AD [75,76,77]. Interestingly, IFN- γ and TNFα not only play a protective role during acute HSV-1 infection, but also during virus reactivation from latency [78]. HSV-1 infected CNS cells also secrete matrix metalloproteases (MMP3, MMP8, MMP9) and chemokines, such as CCL2 (MIP-1α), CCL4 (MIP-1β), CCL5 (RANTES), CXCL9, chemokine (C–X–C motif) ligand 10 (CXCL10), and CX3CL1 [79]. The production of specific cytokines by neurons, astrocytes, and microglial cells after HSV-1 infection is shown in Figure 2.

Cytokines and other molecules produced during inflammation, recruit lymphocytes and myeloid cells to the site of infection through the lymphatics network or across the inflammation-weakened blood–brain barrier (BBB) (Figure 2), and thereby monitor and conquer the invading virus. In the TG of mice latently infected with HSV-1, an influx of lymphocytes, macrophages, and microglial cells around neurons, as well as occasional neuronophagy, was observed. Furthermore, chronic inflammatory foci were present in their brainstem and other brain regions, such as the olfactory bulbs, temporal and parietal areas of the cortex, and leptomeninges [45]. Activation of the glial cells, the major mediators of neuroinflammation [88], and infiltration of the CNS with inflammatory cells, serve to eliminate infected cells and inhibit viral expansion; however, immense and/or persisting inflammation can lead to pathological outcomes. For example, it is inflammation as a defense reaction and result of innate immune responses that can contribute to the secretion of Aβ by glial cells and neurons; however, Aβ accumulation, as well as NFTs formation, can lead to astrocytosis and microgliosis [89], and ultimately activate and deepen the neuroinflammatory state that has a significant share in the progression of AD. This renders inflammation in neurodegenerative diseases the meaning of the double-edged sword [90,91]. Interestingly, HSV-1 evolved specific mechanisms to prevail over the host inflammatory response, e.g., by expression of the proteins ICP0 and virion host shutoff (vhs) blocking IRF3- and IRF7-mediated activation of ISGs [92], and by targeting one of the major viral DNA sensor proteins, IFI16, for rapid proteasomal degradation [70]. Moreover, Hill et al. discovered that HSV-1 infection of primary human neural (HN) cells up-regulated host microRNA-146a (miRNA-146a) associated with proinflammatory signaling and AD (Figure 2), enabling viral evasion from the complement system [87]. Subsequently, a plethora of the host miRNAs playing a role in regulating cell apoptosis, inhibition of viral replication, and, most importantly, antiviral immunity, have been discovered to be deregulated by HSV-1 [41].

In addition to the infiltration of inflammatory cells around blood vessels and white matter necrosis, HSV-1 infection of cotton rats contributed to multifocal CNS demyelination, which, despite subsequent remyelination, resulted in “scars” in the myelin sheaths [93]. Incomplete remyelination can be caused by the transition to a reactive state, as well as by the death of the glial cells responsible for this process, such as astrocytes and microglia [94], and contribute to clinical manifestation in the form of cognitive decline, as myelin impairment can play a significant role in AD pathology and precede Aβ and tau pathologies during the disease [95].

After establishing latency, HSV-1 can reactivate and most often cause vesicles and ulcers in the mucocutaneous sites; however, infrequently, the virus can invade and replicate in the brain. In addition to exceptionally severe, acute encephalitis, HSV-1 can cause mild or asymptomatic subacute illness, also associated with cerebral dysfunction [96]. In adults, it can take the form of manifestations, such as headache, drowsiness, nausea, vomiting, disorientation, photophobia, and weakness [97]. It is believed that mild disease probably exemplifies the more common presentation of HSE; however, less severe symptoms may not be acknowledged, or the patient can recover prior to diagnosis and, thus, many cases can be omitted. However, mild cases of HSE, as well as repeated reactivations of HSV-1, which can also occur in the brain, should be of particular interest, as they can accumulate adverse consequences of inflammation. Whether subacute HSE can contribute to the onset of neurodegenerative disease, such as AD, warrants further investigation.

Moreover, once occurred, HSE can entail long-term, persistent inflammatory processes in the brain. Viral DNA, but no HSV antigens, were found in the patient’s brain ten years after acute onset of HSE [19], indicating that following HSE, HSV-1 can establish latency not only in the TG but also in the CNS cells, and constitute an ember for future brain infections as well as recurrent HSE. Such a possibility was also proposed by Olsson et al. [98]. Interestingly, HSV reactivation in the CNS can appear spontaneously, affect only a small group of cells, and not induce neurological symptoms [45]. Research on the human brain, within the context of atypical/mild/chronic HSV-1 infections, can contribute to an understanding of whether and to what extent molecular and cellular events during such infections promote neurodegeneration. Although they are known to exist, the prevalence of such infections is not underpinned in the population level. Research regarding the long-term consequences of HSV-1 infections of the CNS are scarce, and often hampered by small/selected study groups limited to patients with severe HSV encephalitis [99].

It should be kept in mind that neuronal cells are terminally differentiated and do not produce cellular DNA; therefore, they are deprived of the precursors for HSV-1 DNA synthesis that are encoded by the HSV-1 genes [100]. However, these genes are inessential for viral replication in glial cells multiplying upon brain injury, or in cell culture. During studies on neurodegenerative diseases, increasing attention is being directed towards glial cells triggering neuroinflammatory responses, as neuroinflammation is a composite process orchestrated primarily by various groups of glial cells in CNS, and also peripheral immune cells [101].

## 4. Functions of Astrocytes, Oligodendrocytes, Microglia, and Their Activation

### 4.1. Astrocytes

Astrocytes as the heterogeneous and most numerous cells of the brain, are the major controllers of synaptic activity and plasticity, neuronal network, and cognitive functions [102,103]. Being supportive glial components in neural tissue, they provide an adequate ionic milieu for neurons, support their metabolism, maintain the BBB, regulate blood flow, and clear cell debris, as reviewed by Kim et al. [104]. Furthermore, through participation in the immune response, astrocytes play a critical role in host defence during viral infection, and any alteration in the astrocytic function can contribute to pathological changes in the CNS and neurological complications.

As a consequence of HSV-1 infection, astrocytes undergo a dramatic transformation [57,105]. Such a process, named “reactive astrocytosis”, involves morphological and molecular changes that result in increased astrocyte proliferation, change of cell morphology with a loss of astrocytic projections, and increased levels of proteins, such as glial fibrillary acid protein (GFAP), as shown in Figure 3.

Recently, it was shown that astrocytes exert an increased expression of heparan sulfate proteoglycans—the first binding sites for the virus and, as a consequence, show a greater susceptibility to HSV-1 infection than neurons [58]. The reduction of viable cells and changes in primary human astrocytes’ morphology, from a stellar to globoid shape, was observed after HSV-1 infection [113]. Potent astrocytosis and a dramatical increase in the number of reactive astrocytes were shown in the brain regions damaged by HSV-1 infection [112]. These effects were present both in acute (2 days p.i.) and in chronic (30–60 days p.i.) infections [112]. HSV-1 up-regulates astrocytic secretion of TNFα and IL-6 via the TLR3 pathway [114], as well as induces the production of type I IFNs [16,115]. A recent report showed that HSV-1-infected astrocytes were transiently activated, became hypertrophic, and expressed both pro-inflammatory neurotoxic A1- and anti-inflammatory neuroprotective A2-astrocyte markers [57]. Furthermore, the HSV-1 infection cell protein 0 (ICP0) triggered fibroblast growth factors (FGFs) activity and up-regulated FGF-4, FGF-8, FGF-9, and FGF-15, inducing paracrine neurotrophic signaling in neighboring cells. Furthermore, the up-regulation of TLR2, TLR6, TLR9, MDA5, and DAI, as well as the increased expression of type I IFNs and ISGs, occurred in HSV-1 infected astrocytes [116]. Several days after HSV-1 infection, an increased level of GFAP was observed, indicating the development of astrocytosis [110]. GFAP plays a crucial role in the progress of reactive astrocytosis, in response to viral infections [117]. Reactive astrocytes can provoke the dysfunction of normal astrocytes and affect their response to inflammation [118,119]. The role of astrocytosis in the course of HSE is deliberated. Some studies have reported that astrogliosis exerts beneficial effects, including wound closure, neuronal protection, and BBB repair [120]. However, others have shown that astrocytosis can be harmful, particularly in the context of inflammation [53,121]. Novel evidence showed that astrocytes contribute to and promote HSV-1 infection of neurons by fueling neurons with extracellular ATP. In the absence of functional astrocytes, neuron HSV-1 infection was less efficient; therefore, astrocytes can participate in a productive infection of neurons rather than protecting them [58]. Moreover, reactive astrocytes restrain survival as well as the differentiation of the oligodendrocyte precursor cells [122].

### 4.2. Microglia

Microglia and astrocytes possess immune functions and respond to invading pathogens, by producing soluble mediators that can promote inflammation and leukocyte recruitment across the BBB [123,124,125]. Microglia comprise 10–20% of the glial cells, and being professional CNS macrophages, they sense and internalize extracellular material, cell debris, as well as apoptotic cells, preserving neuronal networks and repairing CNS injuries [126,127]. Microglia are considered the first line of defense in response to HSV-1 infection and are responsible for the release of pro-inflammatory cytokines and chemokines, including type I IFNs, IL-1β, IL-6, TNFα, CXCL10, and C–C motif chemokine ligand 2 (CCL2), and are the major source of inducible nitric oxide synthase (iNOS) [83,128,129], as presented in Figure 2. Importantly, microglia are the main producers of type I IFNs among CNS cells, following HSV-1 infection [130]. The up-regulation of *P2RY12*, *CD68*, *Serpina 3n*, *GFAP*, and *Vim* genes associated with the reactivity of astrocytes and activation of microglia, was recently shown in an HSV-1 infected mouse cortex and hippocampus, conjointly, with the heavy deposition of Aβ aggregates [131]. The up-regulation of *CD68* and *P2RY12* drives microglia into a phagocytic state, suggesting that Aβ aggregates can prime reactive microglia for phagocytosis.

### 4.3. Oligodendrocytes

Oligodendrocytes produce various neurotrophic factors and form myelin sheaths around neurons, taking part in the propagation of potentials along axons [132]. They express various innate immune receptors and modulate immune responses in the brain as well as produce small vesicles—exosomes containing regulatory RNAs and proteins, which play important roles in neurodegenerative disorders. Furthermore, following HSV-1 infection, oligodendrocytes can secrete MVs harboring viral proteins, nucleic acids, or infective virions, thus participating in the viral cycle [51], as shown in Figure 3. Oligodendrocytic HSV-1 infection results in cell death, demyelination, and the loss of neurons [111,112].

During AD, astrocyte proliferation and the transition to a reactive state can be observed. In particular, oligomeric Aβ and hyperphosphorylated tau induce functional astrocyte impairment, resulting in disturbed neuron metabolism, prostrate synaptic activity and plasticity, as well as malfunctioning regional blood supply, as reviewed by Acosta et al. [133]. Furthermore, activated microglia proliferate and concentrate around Aβ plaques, and microglial functioning and responses to Aβ are altered, which results in the secretion of inflammatory factors, exacerbation of tau pathology, and activation of neurotoxic astrocytes, culminating in neuron injury, synapse loss, as well as microglial degeneration [134,135]. Oligodendrocytes and their production of myelin determine the metabolic and trophic surrounding for axons; therefore, they are decisive for the proper functioning of the brain [136]. Their damage can contribute to neurodegeneration and recent data indicate that AD is associated with demyelination, oligodendrocyte dysfunction, as well as the loss of oligodendrocyte–axon communication [137,138]. Deprivation of the myelin sheath can be an initiating step of the early stage AD changes observed before the Aβ and tau pathology [139]. Oligodendrocytes are vulnerable cells, and Aβ, NFTs, oxidative stress, and inflammation induce/aggravate their dysfunction and myelin breakdown, and lead to apoptosis [139,140]. Oligodendrocyte progenitor cells (OPCs), or NG2-glia, regenerate damaged oligodendrocytes; however, they are also disrupted during AD [136].

Adverse and neurotoxic changes occurring in brain cells during HSV-1 infection, can open the door to the development of AD-related pathologies and, importantly, often resemble those observed during the disease. Supporting and enhancing the neuroprotective properties of the glial cells, as well as appropriate components in their responses, have potential employment in a therapeutic approach to dementia and AD in particular [141].

## 5. Oxidative Damage

Reactive oxygen species (ROS) are continuously produced in the mitochondria of the metabolically active CNS cells, and provide an optimal redox state for the activation of transduction pathways involved in the proper regulation of the neuronal activity, glial cells specification, and differentiation [142,143]. However, during mitochondrial dysfunction, when the production of ROS is excessive or exceeds the antioxidant capacity of the cells, oxidative stress can occur [144], as shown in Figure 3. Metabolic processes in neurons trigger ROS production, and astrocytes regulate oxidative stress in the CNS through decomposing and clearing free radicals produced by neurons and other cells. However, under pathological conditions, astrocytes can become a major source of excessive free radicals that can damage neurons and activate microglia, as reviewed by Chen et al. [145]. Activated microglia exhibit a high tolerance to oxidative stress and, in addition, release radicals, such as superoxide and nitric oxide [146], while oligodendrocytes, in contrast, are particularly vulnerable to oxidative stress, high ROS levels especially affect their maturation, myelin sheath formation, and remyelination [143,147].

Extensive scientific evidence links HSV-1 infection with both AD and oxidative stress in CNS cells. During HSV-1 infection, ROS and reactive nitrogen species (RNS) limit viral replication, and regulate host inflammatory and immune responses [148]. A recent study indicates that ROS generation in HSV-1 infected immune cells is NF-κB-dependent, and antioxidant administration raises HSV-1 replication levels in these cells [149]. On the other hand, scientific data show that oxidative stress concurs with HSV-1 replication [150,151]. For instance, the treatment of HSV-1-infected Vero cells with embelin, which possess antioxidant properties, not only reduced the production of H_2_O_2_ and HSV-1-caused oxidative damage, but also inhibited the attachment and penetration of HSV-1 virions, causing the inhibition of infection [152]. However, the antioxidant capability of the host cells can be exceeded by high ROS production, engendering harmful effects. Furthermore, the polyunsaturated fatty acids (PUFA)-rich nervous system is especially prone to oxidative damage, including the autocatalytic lipid peroxidation, during which reactive carbonyl species (RCS) are formed [153]. ROS and RCS prostrate glutathione and other reductants, generate oxidative modifications of nucleic acids and proteins, often modifying their structure and function, which, acting cumulatively, inevitably lead to brain tissue damage and dysfunction. Oxidative injury is an inseparable companion of HSE [154], and is also associated with neurodegenerative diseases. In particular, oxidative stress plays a major role in the pathogenesis of AD [148,155], and its derivatives can represent an early phenomenon of the disease. Increased levels of 8-hydroxyguanine (8-OHG), 8-hydroxyadenine (8-OHA) DNA adducts, and 5,6-diamino-5-formamidopyrimidine were observed in various AD-affected brain regions and hippocampus; in particular, in the early stage of disease [156], while 8-OHG RNA adducts decreased and acrolein/guanosine DNA adducts and 8-OHA RNA adducts increased in the late stage of AD [156,157].

Oxidative damage associated with HSV-1 infection, encompasses primarily lipid peroxidation products’ 4-hydroxy *trans*-2-nonenal (HNE) and acrolein adducts to proteins and 8-OHG, as well as 8-OHA adducts to DNA and RNA, among at least 20 ROS-modified bases adducts. ROS levels were significantly increased in neural P19N 1cells as early as 1 h post infection (p.i.) with HSV-1 [151]. A total of 8-OHG adducts occurred in a large number of neurons, as well as the non-neuronal cells of the TG, brainstem, hypothalamus, and thalamus of mice with an active form of HSV infection [45]. HNE-protein adducts were also observed in latently infected mice. Interestingly, the presence of HNE has also been demonstrated in neurons and non-neuronal cells, in which the presence of HSV-1 proteins or LAT expression in acutely and latently infected mice, respectively, has not been demonstrated; therefore, the damage resulting from HNE attachment was not only limited to HSV-1 infected cells, but also the neighboring ones. Similarly, the presence of 8-OHG has been observed in infected and adjacent cells. The described oxidative damage of neural tissue that occurred during acute and latent HSV-1 infections, was also associated with the apoptosis of the CNS cells, mainly non-neuronal cells. According to multiple HSV-1 reactivations, significantly increased levels of proteins with HNE and 13-HNE adducts were observed in the cortex of HSV-1 infected mice [158]. In particular, the functioning of the two proteins (glucose-regulated protein 78 (GRP78) and collapsin response-mediated protein 2 (CRMP2)) associated with the AD pathophysiology and responsible for stabilizing the microtubules, was impaired.

The above data strongly suggests that HSV-1 infection, and especially recurrent viral reactivation in the brain, can contribute to oxidative damage that can predispose the brain to AD or result in the development of neurodegeneration. Furthermore, HSV-1 infection and oxidative stress alter the lysosome system in the form of increasing the lysosomal load, diminishing the activity of the lysosomal enzymes, or modifying cathepsin maturation, which can be involved in different forms of AD [159]. Oxidative stress during HSV-1 infection, significantly potentiates the intracellular accumulation of Aβ levels and precipitates the abundance of autophagic compartments in SK-N-MC human neuroblastoma cells; therefore, HSV-1 infection together with oxidative damage promote AD neurodegeneration events [160].

## 6. Amyloid Beta Secretion

A widely accepted hypothesis called “amyloid cascade”, designates Aβ accumulation as a factor leading to AD pathology [161,162]. Aβ is generated via proteolytic cleavage of the amyloid precursor protein (APP) by β-secretase (BACE1) and γ-secretase [163]. Until recently, nerve cells were considered the major producers of Aβ; however, the latest scientific findings show that astrocytes can also secrete significant quantities of Aβ [6], and are therefore important cells contributing to cerebral amyloid loading. It is worth noting that amyloid production accompanies astrocytes reactivation, as astrocytic levels of APP, β-, and γ-secretase significantly increase [6]. Astrocytes, however, also act in the opposite manner, by phagocyting and breaking down Aβ. The gravity of this feature is evidenced by the pharmacological ablation of astrocytes in organotypic brain culture slices (OBCSs) from 5XFAD mice, that entailed an increase in Aβ levels, reduced Aβ degradation, as well as reduced the density and size of hippocampal dendritic spines [164], following the ablation of the astrocytic proliferation in APP23/GFAP-TK transgenic mice, which similarly exacerbated the disease pathology in the mouse model of AD [165]. Interestingly, when astrocytes phagocytose Aβ protofibrils, the material may not be degraded, but stored intracellularly, and subsequently secreted from cells in the form of microvesicles containing N-terminally truncated Aβ, which can induce neuronal apoptosis [166], as presented in Figure 4.

Mature oligodendrocytes express various APP isoforms and secrete the 40 and 42 amino acid Aβ species in vitro [179]; however, more research is needed to establish the role of these cells as a source of Aβ, bearing in mind that Aβ peptides are cytotoxic to oligodendrocytes and induce oligodendrocyte death [180,181].

Microglia, being professional phagocytes, are capable of internalizing different forms of Aβ [182,183]. During AD, microglia gather around, interact with Aβ plaques, and become activated through transcriptional and functional reprograming; however, their motility and phagocytic activity become greatly impaired [184]. Interestingly, transient microglia ablation in APP transgenic CD11b-HSVTK mice did not inhibit amyloid plaque formation and maintenance [185]. Additionally, microglia accord to the plaque growth via Aβ clusters phagocytosis and the consecutive release of accumulated Aβ into extracellular space preceding their death [186], as shown in Figure 4. Upon exposure to amyloid, microglia also increase the internalization of amyloid-loaded neurons, even before the amyloid plaque deposition occurs [172]. Dysfunctional microglia can therefore contribute to the aggravation of Aβ deposition and neuronal pathology, before the plaque onset occurs and in the later stages of AD [172,184].

A high number of reactive astrocytes and activated microglia have been found in the vicinity of Aβ plaques, in the brains of the triple transgenic mouse model of AD (3xTg-AD) [169] and people suffering from AD, as reviewed by Fakhoury [69]. Astrocytes and microglia clear and break down amyloid in response to neurodegeneration. However, since microglia also facilitate the convergence of soluble and oligomeric amyloids within plaques into a fibrillar form resistant to degradation, this can result in the loss of the debris-cleaning role of the cells and the development of AD-promoting pathology [187].

A significant increase of Aβ deposits was observed in the brains of HSV-1-infected mice [163,188]. Interestingly, BACE1 and nicastrin (components of γ-secretase) levels increased in HSV-1 infected human neuroblastoma SHSY5Y cells [163]; HSV-1 can also elevate IFN-induced PKR level, leading to the expression of *BACE1*, which is otherwise constitutively inhibited [189]. Moreover, HSV-1 particles interact intracellularly with APP, facilitating viral transport and simultaneously disrupting proper APP transportation and distribution in cells [190], further indicating the possible contribution of HSV-1 to the development of AD pathology. Another significant evidence of the HSV-1 influence on the Aβ deposition in the brain is the demonstration that 90% of Aβ plaques contain HSV-1 nucleic acids, while over 70% of viral DNA was associated with the plaques in the brains of people suffering from AD [191]. HSV-1 is capable of enhancing Aβ aggregation both in cell cultures in vitro as well as during 5XFAD mice infection, and can lead to Aβ nucleation and the growth of amyloid fibrils [192]. In 2019, Ezzat et al. [192] established that HSV-1 binds amyloidogenic peptides to its “protein corona”, a layer of peptides that adhere to the surfaces, thanks to which the formation of amyloid is catalyzed by surface-assisted nucleation. Moreover, increased soluble Aβ (sAβ) can also indirectly contribute to AD, as it belongs to one of the proinflammatory cytokine-induced DAMPs, and, through the activation of the TLR4/TLR7/TLR9 pathway, exacerbates inflammation in the brain [193]. Although the available evidence of HSV-1 presence in the brains of AD patients is not sufficient to confirm an inevitable role of the virus on its own in AD, because HSV-1 also resides in the brains of healthy people [1], the discussed data corroborate that the presence of HSV-1 in the brain can be one of the factors initiating the formation of Aβ plaques, as well as one of the significant contributions leading to the onset of AD.

## 7. Tau Hyperphosphorylation

The anomalous deposition of aggregated proteins in the form of intraneuronal tau filaments is a hallmark of AD, as well as most other neurodegenerative diseases [8]. The neuron-resident tau belongs to the microtubule-associated proteins (MAPs) family, participates in the stabilization and polymerization of microtubules, and thus fosters axonal transport and supports neuronal integrity [194]. Tau and tau-like proteins are also expressed in lower levels in glial cells, such as oligodendrocytes and astrocytes [195,196], where they contribute to the cellular differentiation, formation of myelin sheaths, the outgrowth of processes, and neuron-glia contact [197]. The hyperphosphorylation of tau induces the release of the protein from microtubules [198], the loss of the cytoskeleton, accumulation of tau in the nuclei of neuronal cells, and promotes the formation of aggregates, NFTs [175], as shown in Figure 4. Diminishing axonal stability and hastening neuronal dysfunction supervene on tau hyperphosphorylation [199]; however, such modification does not lead to pathology during hibernation [200] and the fetal development of rodents and humans [201,202,203,204,205,206]. It is speculated that the formation of aggregates from abnormally hyperphosphorylated tau into NFTs, can comprise a protective mechanism by which cells minimize the toxic activity of the abnormal protein [207].

Recently, much scientific attention is directed to the “tau propagation hypothesis”, which assumes that the transmission of tau aggregates between neurons promotes the formation of successive aggregates of tau and AD pathology. The neuron-to-neuron transfer of tau requires release/leakage of the pathological form of tau outside the degenerated cell and its internalization by recipient cells, resulting in the formation of tau aggregates within these cells [208]. Moreover, the increase in the intracellular pathological tau in the brain correlates with the extent of cognitive deficits and the characteristic pattern of tau aggregates spreading during AD, typically the aggregates incrementally accumulate at the cortex of the temporal lobe and propagate to the hippocampus and other parts of the brain [209]. Pathological tau present in glial cells also exerts propagation across the brain, e.g., oligodendroglial tau aggregates spread along white matter tracts, ultimately leading to the loss of oligodendrocytes [210]. In addition to neurons, astrocytes also produce, internalize, and degrade tau, and participate in tau propagation in the brain, also possibly influencing AD progression [177]. Interestingly, glial tau aggregates can supplement neuronal tau aggregates in the same degenerating brain regions [195].

During replication in the nucleus, which is a complex and structured process, HSV-1 recruits numerous nuclear proteins. Although the virus does not require tau for replication in neuronal cells, HSV-1 infection causes tau phosphorylation at serine 202/threonine 205, threonine 212, serine 214, serine 396, and serine 404 [86,211,212], as well as the accumulation of hyperphosphorylated tau in the nuclei of infected neuronal cells [175]. Tau phosphorylation was observed in the cytoplasm of primary adult murine hippocampal neurons already at 24 h of HSV-1 infection [174], and the levels of tau phosphorylated threonine 205, as well as tau cleavage and aggregation increased significantly in the brains of mice with multiple HSV-1 reactivations [86]. The virus enhances the activity of the enzymes responsible for the tau phosphorylation, such as glycogen synthase kinase 3beta (GSK-3β) and protein kinase A (PKA) [213]. Moreover, the pattern of proteins phosphorylated by HSV-1 protein kinase U(S)3 overlaps that of phosphoproteins targeted by PKA [214]. Moreover, HSV-1 infection promotes tau cleavage through caspase-3 activation in murine primary neurons and astrocytes, which increases the kinetics of tau aggregation [215].

Interestingly, it is proposed that tau phosphorylation and accumulation can play a role in antiviral protection as a normal host immune repertoire. Exceeding the critical level of the modified protein and/or its aggregates can redirect their character from shielding to neurotoxic, and lead to AD as to the innate immunity disorder [216]. An adaptation of such a novel perspective on changes during neurodegeneration does, however, require additional observations and experimental data. Furthermore, studies on tau pathology in non-neuronal cells, such as astrocytes and oligodendrocytes, indicate that these cells can be significant players among propitious AD treatment methods.

## 8. Apoptosis and Autophagy

Immense neuronal death, due to apoptosis, is a frequent finding in the brains of people suffering from neurodegenerative diseases, and, in AD, apoptosis entails the extensive death of neurons and glial cells [211,212,217]. It is generally known that viral infections can trigger apoptosis, a programed cell death that plays an important role in viral pathogenesis and host antiviral response [218]. Apoptosis can limit the viral spread; therefore, HSV-1 modulates cellular death and encodes anti-apoptotic virulence factors to evade elimination. On the other hand, HSV-1 also promotes the death of cells that can be detrimental for viral replication [219]. The HSV-1 acute infection of the CNS ineluctably entails neuronal and glial cell death [112,220,221,222], as shown in Figure 4; however, the replicating virus also counteracts neuronal antiviral mechanism, such as autophagy [223].

Apoptosis is modulated by two signaling pathways, extrinsic and intrinsic, and is executed by a family of cysteine proteases known as caspases, recently reviewed by Duarte et al. [15]. There are many studies that show that HSV-1 infection causes neuronal apoptosis and brain disease [220,224,225,226]. Neuronal apoptosis occurred in human HSE brain tissue and cultured human glioblastoma cells infected by HSV-1 [227]. DeBiasi et al. [220] observed apoptotic neurons and glia in brain tissue sections of patients with acute HSE, indicating that HSV-1 infection can directly cause apoptosis of the BBB components. He et al. [228] indicated that HSV-1 infection triggers apoptosis as a consequence of BBB damage, which was associated with a GM130-mediated Golgi stress response, and HSV-1 neuronal infection entails Golgi apparatus fragmentation [229]. Recently, it was shown that the HSV-1 infection of mice causes hippocampal damage and neuronal apoptosis, which is related to the downregulation of the suppressor of cytokine signaling 2 (SOCS2) and SOCS3, and to increased hippocampal expression of inflammatory cytokines, such as TNFα, IL-1β, IL-6, and IFN-α/β [230]. In the cultured neurons, HSV-1 infection decreased the expression of the dendritic postsynaptic density scaffolding proteins, such as postsynaptic density protein 95 (PSD-95), Drebrin, and CaMKIIb, and induced extensive loss of dendritic spines and retraction of secondary dendrites, as well as entailed unresponsiveness to glutamate stimulation, culminating in the functional deregulation of neurons [231]. Latterly, Doll et al. provided evidence that sensory neurons undergo apoptosis as a result of HSV reactivation in mice [232]. In this study, dead neurons were cleared by Iba1+ cells, which can play a role in preventing the damage and protecting neighboring neurons. HSV-1 infection also induces microglial apoptosis. Expression of the apoptotic genes of caspase-2, caspase-3, Cide-B, and Dsip1 were increased; however, *Tnfrsf12a* and *RipK2* were down-regulated in microglia after HSV-1 infection [233].

HSV-1 modulates the apoptotic pathway through the expression of viral immediate early (IE) genes, e.g., *ICP0* acts as an activator of apoptosis. The expression of ICP0 alone was necessary and sufficient to trigger apoptosis during HEp-2 cells infection by HSV-1 [234]. A new study presented by Mangold et al. [235] showed that HSV-1 modulates viral gene expression and protein levels, depending on the viral strain in infected human neuronal cells. In turn, infected neurons change the response pathways to different HSV-1 strains and activate genes involved in death receptor signaling and retinoic acid-mediated apoptosis signaling. Furthermore, the changes included pathways that regulate neuronal cell adhesion, migration, and cytoskeletal rearrangement (pathways associated with integrin signaling, integrin-linked kinase (ILK), ephrin B-, and ephrin receptor-signaling), as well as the regulation of neuronal adherens junction components in response to particular HSV-1 strains [235].

HSVs can also modulate autophagy [236]. The host defense against HSV-1 infection, involving autophagy, relies on fencing off the cellular synthesis of virus proteins, reached by phosphorylation of the eIF2α [237,238]. eIF2α phosphorylation promotes the induction of autophagy in HSV-1-infected neurons [27]. Autophagy can also be elicited by recognition of viral DNA by the cGAS DNA sensor and beclin 1 (BECN1), or by the stimulator of interferon genes (STING). These pathways lead to the delivery of viral DNA to autophagosomes [239,240]. Furthermore, HSV-1 infection in murine TG neurons induced autophagosome clusters also in non-infected neurons lacking detectable viral protein expression [241]. HSV-1 utilizes cellular mechanisms to inhibit autophagy through virus neurovirulence proteins: US11 and ICP34.5 [27]. US11 interacts with PKR and inhibits eIF2α phosphorylation [242]. ICP34.5 binds to BECN1 and inhibits its autophagic function. A previous report showed that the mutant HSV-1 lacking the BECN1-binding domain of ICP34.5 did not inhibit autophagy in neurons [223].

Significantly, autophagy can also be exploited by viruses to enhance their multiplication or persistence during latency. Recently, it was reported that the transient induction of autophagy by HSV-1 in human monocytic THP-1 cells appeared to have a proviral role [243].

It is worth to mention that the hosts’ exploitation of autophagy against HSV-1 is cell type-specific. For example, autophagy was critical for viral control in cultured primary neurons, while it was dispensable in fibroblasts in vitro [244,245]. This difference was also observed in vivo, autophagy was involved in the antiviral response in neurons, but not in epithelial cells [245]. Interestingly, HSV-1 infection entails the accumulation of intracellular autophagosomes and Aβ affluence in autophagic regions in human neuroblastoma cells [246], indicating a possible multifaceted role for HSV-1 in the development of AD.

## 9. Neuronal and Glial Cell Injury and Loss Combined with Advanced Age

Advanced age favors the development of processes that can affect neurodegeneration and, at the same time, the possibility of having HSV-1 infection and virus in the brain increases with the length of life. Notwithstanding the fact that changes observed in the brain during aging resemble AD pathology, the disease is not synonymous with accelerated brain aging [247]. During AD, neuronal loss occurs with the distinction of regional selectivity, principally, in the hippocampus and neocortex [248,249]. Research indicates that neuron death in brain aging, however present, may not necessarily precipitate the age-related deterioration of hippocampal and neocortical functions [250]. According to changes in the hippocampal subfield structure, AD-related processes can be qualitatively different from those occurring during the normal aging of the brain [248]. However, Avramopoulos et al. [251] found a particularly significant overlap between changes of gene expression with age and changes in AD. Furthermore, the group observed that up-regulation of genes involved in the inflammation and regulation of transcription as well as down-regulation of genes associated with neuronal functions eventuated in the same direction. Consequently, although structural and genetic alterations observed in healthy-aging brains can be divergent from those appearing in AD, advancing age is a risk factor for AD and age-related changes and can augment the likelihood of AD development [251,252]. Common characteristics of brain aging encompass changes in neuron and brain volume, alterations in dendritic complexity and neurotransmission, and accumulation of neurotoxic proteins [67]. These processes are suspected to contribute, at least in part, to the neurodegeneration and cognitive impairment related to advanced age, although the grounds of many known brain changes related to aging still remain indefinite.

HSV-1 DNA is present in high frequency in the brains of elderly people, compared to the brains of children and young people in which it occurs only in a very small proportion [1,191,253]. HSV genetic material has been detected in 35% of normal human brains studied, or in the brain tissue of 54% of humans free of clinical signs of HSV-1 infection, suggesting that the virus can establish latency in the brain without severe encephalitic sequelae [254,255]. It has been proposed that the virus reaches the brain in elderly people because the immune system is aging and declines with age [1], while viral reactivations following stimuli, such as stress or immunosuppression, can lead to mild encephalitis and can be attributable to neuronal damage and affect the progression of brain pathogenesis [253,256].

HSV-1 readily infects neurons, astrocytes, microglia, and oligodendrocytes in the brain, neuronal, and glial destruction is observed in the acute stage and during the first month following HSV-1 encephalitis in humans [257]. During encephalitis induced by acute HSV-1 infection, neuronal loss caused by necroptosis and apoptosis, particularly in the temporal and frontal lobes of the human brain were observed [15]. On the second day of HSV-1 infection in mice, focal moderate cortical necrosis and myelin swelling were present in the brain, while from 7–10 days p.i., neuronal degeneration and deficits appeared in severely necrotic areas, and astrocyte/oligodendrocyte nuclei were substituted by karryorhectic figures indicating cell death [112]. Astrocytes of mice infected with HSV-1 by the corneal route undergo active degeneration, and animals exhibit a complete loss of astrocytes in the trigeminal root entry zone already 6 days post virus inoculation [258]. Additionally, moderate focal-to-extensive astrogliosis developed in the brains of animals with subchronic (14–60 days p.i.) HSV-1 infection, characterized by the presence of reactive astrocytes with elongated, thickened, and branching processes, which gather in the areas with a lack of neurons. Brains of animals 30 or 60 days p.i., were visibly reduced in size, and an evident neuronal loss in the hippocampus and loss of as much as 40% of the temporal-occipital cortex were observed. The surviving animals, 30 and 60 days p.i., exhibited behavioral defects and long-term memory deficits. HSV-1 infection of murine neuronal cultures led to a shortening of axons and dendrites after 8 h p.i., reduced neuronal viability to 40% and axonal length to 20% in 18 h p.i., with regard to uninfected control cells, and induced cytoskeletal reduction and retraction, which together resulted in axonal injury, neurite damage, and neuronal death [226]. Moreover, the immense destruction of the nerve terminals was observed following HSV inoculation in the neostriatum of rats, which probably elicited a significant decrease in tyrosine hydroxylase (TH) and glutamine acid decarboxylase (GAD) in the striatum and substantia nigra, and choline acetyltransferase (ChAc) in the ipsilateral striatum [259]. Dopaminergic hypofunction was also observed as a repercussion of the acute HSV brain infection in rabbits [260]. Damage through loss of neurons and cholinergic phenotype to the cholinergic system, which is involved in memory and learning, is considered to be among the earliest events during AD etiology [89], while the loss of dopaminergic neurons pertains to cognitive decline symptoms in a mouse model of AD [261], and structural alterations of the dopaminergic system strongly condition behavioral symptomatology in AD patients, as reviewed by D’Amelio et al. [262].

Interestingly, latent HSV infection in mice also led to the injury of neurons, resulting in the decrease of neuronal parameters in the TG, such as cell diameter, nucleus diameter, the density of cells, and their number, indicating that latent HSV-1 infection in the brain is associated with progressive neuronal pathology [2]. The lytic, quiescent HSV cycle in iPSC-derived glutamatergic human neurons altered cellular function [263], and persistent HSV-1 infection of human and murine neuronal cultures, as well as murine brains, up-regulated *Arc* expression and deteriorated protein activity in maintaining neuronal morphology, synaptic plasticity, and formation of memory [264].

Succinctly, HSV-1 infection in the brain not only instantly affects the physiology of neurons, astrocytes, microglia, and oligodendrocytes, significantly altering their protein levels and morphology [265], but also leads to heavy cell damage and death. Interestingly, the virus exploits molecular strategies to evade host cell death through suppression of both host cell death pathways for the benefit of viral replication [228]; yet, the entry and spread of HSV-1 in the CNS leads to severe and long-term brain damage. Neuroanatomical HSV-1 tropism not only comprises TG or DRG, but also cingulate gyrus, orbitofrontal, insular, and mesial temporal lobe regions of the cerebral cortex [3], including, in particular, the hippocampus, which exhibits a high level of HSV-1 receptors [266]. AD is considered hippocampal-related brain disorder, and cerebral cortex regions that are damaged during HSE correspond to areas in which changes are observed in AD patients’ brains [3].

## 10. Genetic Studies—Future Line of Investigation

The immune system of the majority of humans who have acute/latent HSV-1 infection, is capable of suppressing the virus from entering the CNS; however, a monogenic inborn error of immunity can significantly predispose a man to a neurologic emergency, such as HSE.

Mutations in *TLR3*, genes of the components of the TLR3 pathway, or other genetic deficiencies that lead to the diminished IFN production, can result in HSE among HSV-1 infected patients [267,268,269,270,271,272,273,274,275,276,277,278,279,280,281,282]. Furthermore, mutations in the DBR1, the protein responsible for dsRNA binding, resulted in HSV-1 as well as influenza virus and norovirus presence in the brainstem and brainstem HSE [283]. Recently, rare deleterious variants of *SNORA31* were found in 5 HSE patients [284]. *SNORA31*, one of the most conserved small nucleolar RNAs (snoRNAs), directs the uridine to pseudouridine isomerization in small nuclear and ribosomal RNA; therefore, a lack of functional snoRNA31 outlines novel mechanism underlying HSE. Since TLR3 is required for innate immune responses to HSV-1 in neurons and other CNS cells, such as oligodendrocytes and astrocytes [279,285], deficiencies in TLR3 and other proteins associated with the proper receptor functioning, as well as IFN induction in the brain, also possibly contribute to increased susceptibility to HSE [286,287]; however, this requires further examination.

Lately, mutations in the optineurin (*OPTN*), which determines the selective degradation of the two HSV-1 proteins, VP16 and gB by autophagy, were found to result in increased virus multiplication in the brains of mice, leading to the death of neurons and culminating in accelerated neurodegeneration [288]. *OPTN* deficiency was also associated with increased expression of the proinflammatory markers, IFN-γ, and fewer CD3^+^4^+^ and CD3^+^8^+^ cells in brainstems of mice encountering HSV-1 infection.

Scientific evidence incrementally converges on confirming the hypothesis of HSV-1 being a causative factor of neurodegenerative diseases; however, a question arises whether the HSV-1 ingress to the host brain and subsequent detrimental activity of the virus primarily ensues from the patient’s genetic background. The genetic basis of determining protective immune response to the HSV-1 infection in the brain points to CNS-specific contributions of the TLR3-dependent IFN-α/β- and IFN-λ-mediated immunity, as well as DBR1 and SNORA31 antiviral mechanisms [283,284,289,290]. Studying the collaboration between genetic paucities in the host defense and the emergence of clinical symptoms of HSV-1 brain infection in the form of acute HSE, can lay the novel groundwork for investigating the association of AD onset with former HSV-1 invasion in the brain, resulting from the incomplete or impaired antiviral response.

## 11. Conclusions

HSV-1 infection leads to increasing pathologies in the brain, which can be associated with the emergence of neurodegenerative diseases among people [2]. Both viral latency and viral reactivation can contribute to the degeneration of neurons and their loss, which is the trigger point for the clinical emergence as well as propagation of dementia. Following reactivation, HSV-1 can shed asymptomatically or entail either acute or chronic disease; thus, the infection can cause a broad variety of severities [256]. Many of the sequelae of HSV-1 infection in the brain provide a route to the loss of glial neuroprotection. This article summarizes the most important changes in brain glial cells, following HSV-1 infection and their consequences. Activation of the glial cells, oxidative damage, Aβ secretion, tau hyperphosphorylation, secretion of inflammatory factors, and cell loss in certain regions of the brain are unchanging or similar to the alterations observed in AD; therefore, they can lead to the development of neurodegenerative pathologies promoting AD onset and/or progression.

Although the transition of glial cells into a reactive state, the production of ROS, Aβ, pro-inflammatory agents, and initiation of apoptosis are directed at the damage of the infected cells, they can exert a profound effect on brain functioning. Importantly, the discussed alterations can induce and/or augment each other. For example, the release of inflammatory agents stimulates Aβ production in astrocytes, while Aβ elicits oxidative stress in these cells. Furthermore, soluble tau oligomers can be secreted into the extracellular environment and contribute, independently or in concert with Aβ, to synaptic dysfunction and the ensuing memory loss [291]. Furthermore, Aβ/NF-κB interaction in astrocytes can play a central role in the inflammatory and oxidative stress changes present in the brains of AD patients [292]. The alterations emerging in the CNS cells following HSV-1 infection were collectively presented in Figure 5.

The IFN system is the immunological circuit that plays an explicit role in host defense against viral infections, by limiting virus replication and establishing an overall anti-viral state [35]. Herpes simplex viruses downregulate IFN responses and evade the immune system; therefore, deficiencies in molecular pathways leading to IFN production can sensitize people to HSV-1 brain infections and their long-term clinical consequences.

In the last decade, prominent attention in the research field of AD has been directed towards glial cells in the brain, as their impaired functioning is of substantial importance in AD progression and at the same time provides a significant goal for therapies. Still, many of the molecular mechanisms underlying the pathologies associated with HSV-1 infection observed in glial cells require careful elucidation. These cells, through their neuroprotective and neurorestorative behavior, appear to be key players in the fight to suppress AD.

## Figures and Tables

**Figure 1 ijms-23-00242-f001:**
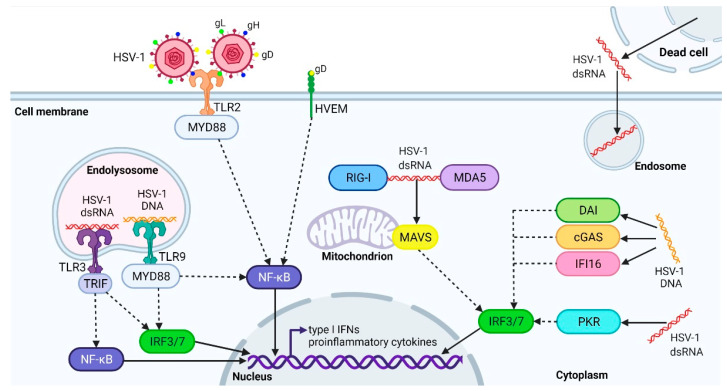
HSV-1 detection by innate immunity. The innate immune system recognizes HSV-1 using different PRRs. Viral glycoproteins H and L are sensed by TLR2, while soluble gD activates HVEM. Viral DNA is recognized by TLR9 localized in the endolysosome or activates DAI, cGAS, and IFI16 localized in the cytoplasm. Viral dsRNA, a replication byproduct, enters the cells from the extracellular environment following the breakdown of infected cells. The dsRNA activates TLR3 in the endolysosome or cytoplasmic receptors, RIG-I, and MDA5, as well as PKR. RIG-I and MDA5 signal via the adaptor mitochondrial antiviral-signaling protein (MAVS) localized at the mitochondrion. The appropriate adaptor proteins are used to govern TLRs signaling pathways: TLR3 signaling utilizes TRIF while TLR2 and TLR9 exploit MYD88. The execution of signaling pathways, shown by the dashed lines, leads to the activation of transcription factors, such as the interferon regulatory factor 3 (IRF3), IRF7 and nuclear factor kappa B (NF-κB), which enter the nucleus and trigger the expression of proinflammatory cytokines and type I IFNs.

**Figure 2 ijms-23-00242-f002:**
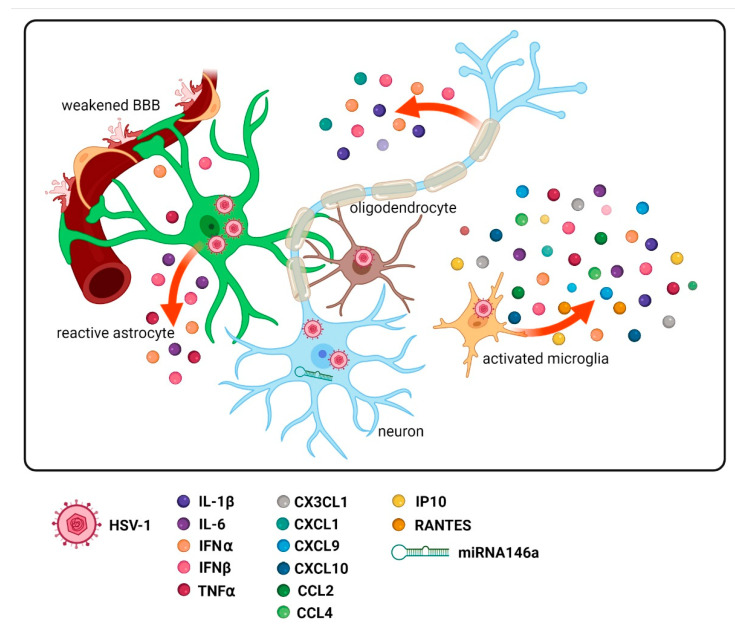
Induction of specific cytokines and chemokines secretion by CNS cells following HSV-1 infection. After HSV-1 entry into the brain, neurons, as well as glial cells, become infected. The BBB becomes weakened [80] and HSV-1 infection leads to the activation of astrocytes, which produce pro-inflammatory cytokines, such as IFN-α, IFN-β, TNFα, and IL-6 [81,82]. Activated microglia secrete IFN-α, IFN-β, IL-1β, IL-6, TNFα, IP-10, CXCL10, CCL2, CCL4, CX3CL1, CXCL9, and CCL5 (RANTES) [83]. Infected neurons produce IFN-α, IFN-β, IL-1β, CXCL1, and CXCL10 [49,84,85,86]. Furthermore, neurons exert deregulated expression of miRNAs, e.g., up-regulated miRNA-146a [87].

**Figure 3 ijms-23-00242-f003:**
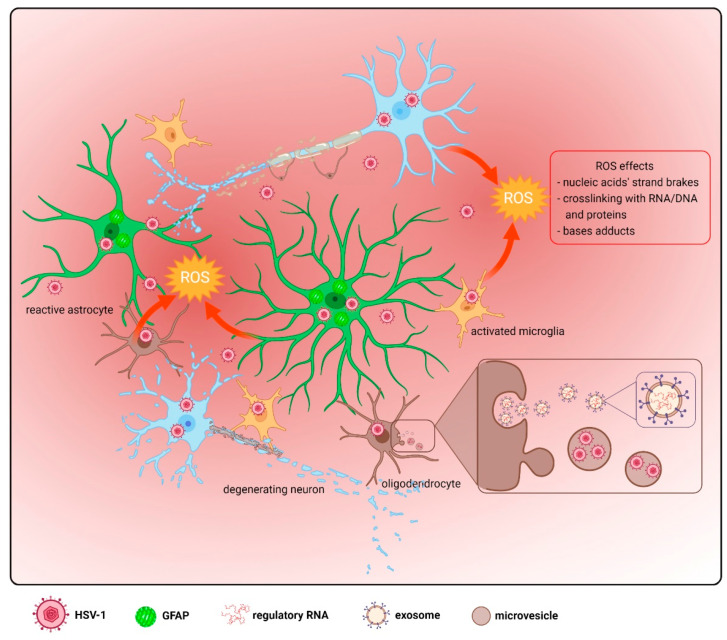
Generation of ROS by the CNS cells following HSV-1 infection. HSV-1-infected neurons undergo oxidative stress and generate excess ROS; however, glial cells are also a significant source of these reactive molecules [106]. ROS radicals compromise the DNA causing strand brakes, crosslink bases of nucleic acids or proteins, or modify DNA bases through the induction of adducts [107,108,109], as shown in the red box. Astrocytes exert an increased level of GFAP following infection, which indicates the development of astrocytosis [110]. Infected oligodendrocytes secrete microvesicles (MVs) containing viral proteins, nucleic acids, or infective virions [51], and produce exosomes containing regulatory RNAs and proteins, as shown in the brown box. Oligodendrocytic HSV-1 infection results in cell death, demyelination, and loss of neurons [111,112].

**Figure 4 ijms-23-00242-f004:**
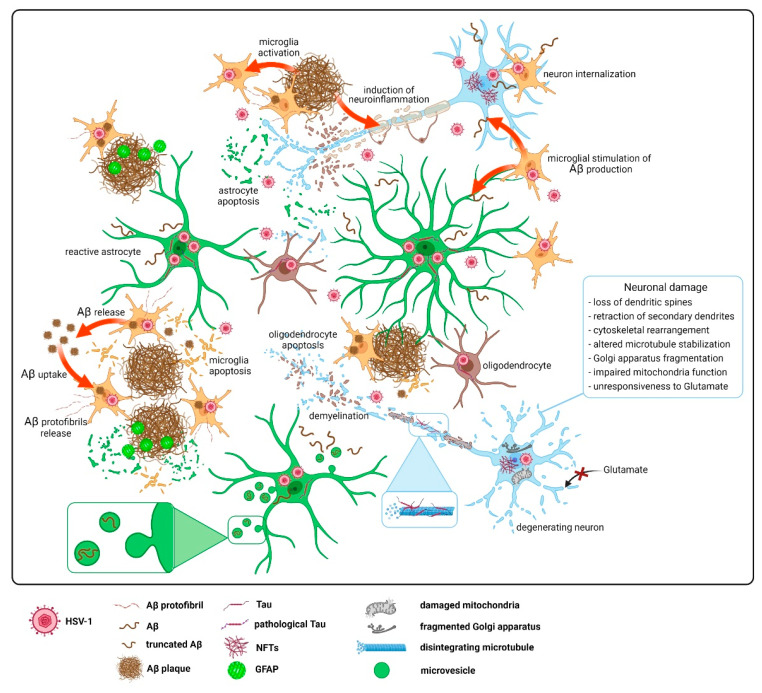
Engulfment and production of amyloid β, accumulation of hyperphosphorylated tau in neurons and glial cells, and death of the CNS cells after HSV-1 infection. Reactive astrocytes can internalize or secrete Aβ, particularly in the form of N-terminally truncated Aβ in MVs, and contribute to neuronal apoptosis [166]. Microglia also clear amyloid plaques or secrete Aβ neurotoxic forms [167,168]. Astrocytes and microglia often encircle Aβ plaques [6,169]. Astrocytes surrounding Aβ plaques can die, leaving an area rich in amyloid and GFAP, and activated microglia can stimulate CNS cells to produce Aβ [170]. In turn, extracellular Aβ induces neuroinflammation and activates microglia [171]. Aβ-activated microglia internalize dead and dying neurons as well as other stressed live cells [172,173]. Demyelinating lesions in the CNS occur following the death of oligodendrocytes after HSV-1 infection [111]. Furthermore, the virus elicits the hyperphosphorylation of tau and accumulation of the protein in the nuclei of neurons and neuronal cells [174,175]. Tau aggregates can be released from neurons [176] and astrocytes, and microglia can internalize the proteins and contribute to their spread [177,178].

**Figure 5 ijms-23-00242-f005:**
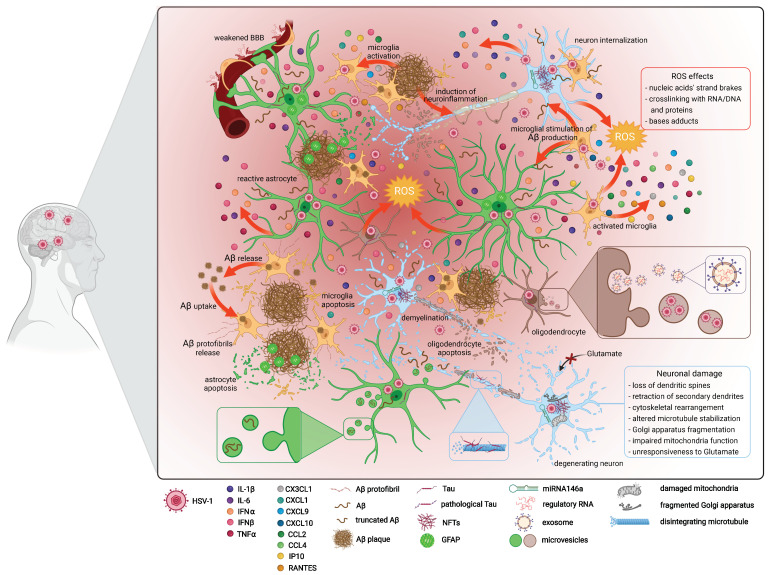
Molecular and cellular events in brain neurons and glial cells after HSV-1 infection. Following HSV-1 entry into the brain, neurons as well as astrocytes, microglia, and oligodendrocytes can become infected. HSV-1 infection can entail Aβ accumulation and the formation of amyloid plaque structures in a 3D bioengineered brain model [293]. After infection, astrocytes become reactive, undergo morphological changes, e.g., loose projections, exert increased levels of GFAP, and secrete pro-inflammatory cytokines, such as IFN-α, IFN-β, TNFα, and IL-6 [81,116]. Astrocyte endfeet are structural components of the blood–brain barrier BBB [53,82], whose structure is impaired during viral infection [80]. Reactive astrocytes often surround Aβ plaques [169]. Following astrocyte death, there remains an area in the vicinity of Aβ plaques containing a significant level of GFAP. Astrocytes have the ability to internalize as well as secrete Aβ [6]. In particular, N-terminally truncated Aβ can be generated and secreted by astrocytes in MVs into the extracellular environment and contribute to the apoptosis of neurons [166]. Furthermore, Aβ in the extracellular milieu induces neuroinflammation, an early event in neurodegeneration [294], and activates microglia [171]. Microglia can phagocyte dead and dying neurons, as well as stressed live cells, their processes and synapses [295], and Aβ-activated microglia increase the internalization of neurites and can induce neuronal death [172,173]. Microglia gather around and clear amyloid plaques [167]; however, they also confine larger Aβ deposits in the plaques [134]. Furthermore, the cells can convert Aβ into neurotoxic pre-fibrillar forms that are trafficked and released in MVs [168]. Additionally, activated microglia can stimulate CNS cells to up-regulate the production of Aβ [170]. In response to HSV-1 infection, activated microglia secrete IFN-α, IFN-β, IL-1β, IL-6, TNFα, IP-10, CXCL10, CCL2, CCL4, CX3CL1, CXCL9, and CCL5 (RANTES) [83]. Oligodendrocytes can support HSV-1 spread by secreting MVs with the infectious virus [51]; however, the death of the cells occurs following infection [15], and HSV-1 induces multifocal demyelinating lesions in the CNS [111]. Oligodendrocytes produce exosomes with regulatory RNA and can send functional molecules to neurons, affecting their properties [296]. HSV-1 influences neuronal physiology, and induces structural disassembly and functional deregulation, as shown in the blue box [231]. Infected neurons largely secrete IFN-α, IFN-β, IL-1β, CXCL10, and Aβ peptides [49,84,85,297]. Furthermore, following HSV-1 infection, neural cells show up-regulated miRNA-146a [87] as well as deregulated expression on other miRNAs [41]. Neuroinflammatory cytokines lead to elevated Aβ concentrations in the brain [298]. HSV-1 infection has been shown to induce complex hyperphosphorylation of tau and nuclear accumulation of hyperphosphorylated tau in neurons and neuronal cells [174,175]. Accumulation of abnormally phosphorylated tau has been demonstrated to precede the formation of NFTs during AD [299]. Tau aggregates are deposited within neurons; however, tau monomers, as well as aggregates, can be released from these cells [176]. Astrocytes, as well as microglia, can internalize tau and contribute to tau spread [177,178]. Neurons, as well as glial cells, can be the source of ROS, which cause significant cellular damage, as shown in the yellow box.

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
