# Peer review of "Disrupting Neurons and Glial Cells Oneness in the Brain—The Possible Causal Role of Herpes Simplex Virus Type 1 (HSV-1) in Alzheimer’s Disease"

_ijms, 2021, doi:10.3390/ijms23010242_

Round 1

Reviewer 1 Report

The work by Mielcarska et al. provides a comprehensive and detailed overview of the role of herpes simplex virus type 1 (HSV-1) as a contributing factor to Alzheimer’s disease (AD), focusing on s the pathological alterations in the central nervous system (CNS) cells that occur following HSV-1 infection. In my opinion, the topic is relevant, and the manuscript is overall well-written and organized. References are appropriate and updated. My only suggestion is to increase the number of figures and/or tables. Indeed, paragraphs 5-8, even though very well written and detailed, are quite difficult to follow; some schematic representations would really help the readers to better acquire the data reported in the Review.

Author Response

We thank the Reviewer for a thorough reading of our manuscript and appreciation of the quality of our work. As suggested by the Reviewer, we increased the number of figures and we think that the Reviewer's recommendation has improved our work. We hope that corrections made to the manuscript will meet the Reviewer’s expectations. The revisions made to the manuscript are described below and are also highlighted in the manuscript in yellow color.

The work by Mielcarska et al. provides a comprehensive and detailed overview of the role of herpes simplex virus type 1 (HSV-1) as a contributing factor to Alzheimer’s disease (AD), focusing on s the pathological alterations in the central nervous system (CNS) cells that occur following HSV-1 infection. In my opinion, the topic is relevant, and the manuscript is overall well-written and organized. References are appropriate and updated. 

Response:

We thank Reviewer for providing a constructive feedback and finding the topic of our article interesting.

My only suggestion is to increase the number of figures and/or tables. Indeed, paragraphs 5-8, even though very well written and detailed, are quite difficult to follow; some schematic representations would really help the readers to better acquire the data reported in the Review.

Response:

We have revised the manuscript text and inserted four figures.
Figure 1 describes innate immune response following HSV-1 infection as described in paragraph 2 (lines 111-123).

Figure 2 illustrates the production of specific cytokines and chemokines after HSV-1 infection (lines 184-192).

Figure 3 shows the production of ROS by the CNS cells following HSV-1 infection (lines 277-287).

Figure 4 demonstrates the engulfment and production of amyloid β, accumulation of hyperphosphorylated tau in neurons and glial cells, and death of the CNS cells after HSV-1 infection (lines 456-469).

Figure 5 summarizes figures 2-4 and illustrates the alterations in the CNS cells following infection with HSV-1 presented previously (lines 787-823).

We have also included new abbreviations (lines 882, 888, 893) and decided to withdraw the term "glial scar", as the word "scar" should remain restricted only to non-neural stromal cells and fibrotic extracellular matrix in line with terminology in other tissues (Sofroniew MV, Trends Immunol 2020).

Reviewer 2 Report

  1. The Manuscript is clear and interesting. Since Figure 1 included a lot of information and events in the brain, readers should be feel some complicate. To further improve it, Figure 1 need a simplification for specific stage or event in the brain. One suggestion, HSV-1 infection or entry into the brain (neuron, astrocytes, microglia, and oligodendrocyte and so on) at Figure 1. After infection events are at Figure 2. Neuronal damage and ROS effect are Figure 3.
  1. Section 4 (Lane 273); Section 4 need a subsection for cell types. For example, 4-1 XXXXX, 4-2 XXXXX and so on.

Author Response

We thank the Reviewer for reading our work and for the constructive review. We revised the manuscript according to the Reviewer's recommendations and increased the number of figures. We think that the Reviewer's suggestions have improved our work and hope that corrections made to the manuscript will meet the Reviewer’s expectations. The revisions made to the manuscript are described here and are also highlighted in the manuscript with yellow color.

1. The Manuscript is clear and interesting.

Response:

We thank the Reviewer for appreciating our work.

Since Figure 1 included a lot of information and events in the brain, readers should be feel some complicate. To further improve it, Figure 1 need a simplification for specific stage or event in the brain. One suggestion, HSV-1 infection or entry into the brain (neuron, astrocytes, microglia, and oligodendrocyte and so on) at Figure 1. After infection events are at Figure 2. Neuronal damage and ROS effect are Figure 3.

Response:

As suggested by the Reviewer, we inserted additional figures. The new figure order is presented as described below:

Figure 1 describes innate immune response following HSV-1 infection as described in paragraph 2 (lines 111-123).

Figure 2 illustrates the production of specific cytokines and chemokines after HSV-1 infection (lines 184-192).

Figure 3 shows the production of ROS by the CNS cells following HSV-1 infection (lines 277-287).

Figure 4 demonstrates the engulfment and production of amyloid β, accumulation of hyperphosphorylated tau in neurons and glial cells, and death of the CNS cells after HSV-1 infection (lines 456-469).

Figure 5 summarizes figures 2-4 and illustrates the alterations in the CNS cells following infection with HSV-1 presented previously (lines 787-823).

2. Section 4 (Lane 273); Section 4 need a subsection for cell types. For example, 4-1 XXXXX, 4-2 XXXXX and so on.

Response:

We thank the Reviewer for the remark, we inserted appropriate subsections for cell types in the section 4.

We have also included new abbreviations (lines 882, 888, 893) and decided to withdraw the term "glial scar", as the word "scar" should remain restricted only to non-neural stromal cells and fibrotic extracellular matrix in line with terminology in other tissues (Sofroniew MV, Trends Immunol 2020).